# Encapsulation in Oxygen-Loaded Nanobubbles Enhances the Antimicrobial Effectiveness of Photoactivated Curcumin

**DOI:** 10.3390/ijms242115595

**Published:** 2023-10-26

**Authors:** Zunaira Munir, Chiara Molinar, Giuliana Banche, Monica Argenziano, Greta Magnano, Lorenza Cavallo, Narcisa Mandras, Roberta Cavalli, Caterina Guiot

**Affiliations:** 1Department of Neurosciences, University of Turin, 10125 Torino, Italy; zunaira.munir@unito.it (Z.M.); caterina.guiot@unito.it (C.G.); 2Department of Drug Sciences and Technologies, University of Turin, 10125 Torino, Italy; chiara.molinar@unito.it (C.M.); monica.argenziano@unito.it (M.A.); greta.magnano@unito.it (G.M.); roberta.cavalli@unito.it (R.C.); 3Department of Public Health and Pediatric Sciences, University of Turin, 10126 Torino, Italy; giuliana.banche@unito.it (G.B.); lorenza.cavallo@unito.it (L.C.)

**Keywords:** nanobubbles, curcumin, oxygen, photoactivation, antibiotic resistance

## Abstract

In both healthcare and agriculture, antibiotic resistance is an alarming issue. Biocompatible and biodegradable ingredients (e.g., curcumin) are given priority in “green” criteria supported by the Next Generation EU platform. The solubility and stability of curcumin would be significantly improved if it were enclosed in nanobubbles (NB), and photoactivation with the correct wavelength of light can increase its antibacterial efficacy. A continuous release of curcumin over a prolonged period was provided by using innovative chitosan-shelled carriers, i.e., curcumin-containing nanobubbles (Curc-CS-NBs) and oxygen-loaded curcumin-containing nanobubbles (Curc-Oxy-CS-NBs). The results demonstrated that after photoactivation, both types of NBs exhibited increased effectiveness. For *Staphylococcus aureus*, the minimum inhibitory concentration (MIC) for Curc-CS-NBs remained at 46 µg/mL following photodynamic activation, whereas it drastically dropped to 12 µg/mL for Curc-Oxy-CS-NBs. *Enterococcus faecalis* shows a decreased MIC for Curc-CS-NB and Curc-Oxy-CS-NB (23 and 46 µg/mL, respectively). All bacterial strains were more effectively killed by NBs that had both oxygen and LED irradiation. A combination of Curc-Oxy-CS-NB and photodynamic stimulation led to a killing of microorganisms due to ROS-induced bacterial membrane leakage. This approach was particularly effective against *Escherichia coli*. In conclusion, this work shows that Curc-CS-NBs and Curc-Oxy-CS-exhibit extremely powerful antibacterial properties and represent a potential strategy to prevent antibiotic resistance and encourage the use of eco-friendly substitutes in agriculture and healthcare.

## 1. Introduction

Antibiotic resistance and the growth of nosocomial infections, which notably afflict elderly people, are the main causes of the bacterial infections that represent an increasing threat to public health. In response, a number of researchers are investigating the potential of organic, natural substances and the use of non-ionizing radiation, such as visible light, to activate natural photosensitizers that can produce reactive oxygen species (ROS) and locally released substances [1,2,3].

Several studies have revealed that curcumin exhibited activity against several bacterial strains. Eren et al. (2020) synthesized a new bone cement composite with curcumin, methyl methacrylate, hydroxyapatite, and Ag+-montmorillonite. The data obtained by the authors confirmed that, in the presence of curcumin, a strong antibacterial character is obtained against both Gram-negative and Gram-positive bacteria [4].

Food contamination has already been the subject of promising microbial photodynamic inactivation (PDI) applications [5]. Our research team has also studied the use of photoactivated curcumin to prevent bacterial fruit contamination [6,7]. This strategy is also promising for reducing the risk of infections during wound healing and with bedsores [8,9,10,11]. A similar approach could potentially be used for many human bacterial diseases that occur on accessible mucosal surfaces, such as the skin, where staphylococci cause atopic dermatitis, if it were proven to be entirely biocompatible [12]. Other possible applications for which photoactivated curcumin could be used include oral mucosae, which can cause dentinal tubules diseases [13], vaginal mucosae, which can be a source for bacterial migration to the urinary tracts [14,15], and, possibly, endophthalmitis [16].

To increase the effectiveness of antibacterial drugs, researchers have investigated the inclusion of photosensitizers into nanovectors [17,18,19,20]. A further strategy is to induce continuous and prolonged release of the photosensitizer using specific nanocarriers that are able to adhere to bacterial walls to locally increase the efficiency of PDI. One such instance is the use of nanovectors covered with bioadhesive polymers, such as chitosan, a well-known mucoadhesive agent. 

Previous studies carried out on nanocurcumin have shown that the MIC of nanocurcumin for *Staphylococcus aureus*, *Bacillus subtilis*, *Escherichia coli*, and *Pseudomonas aeruginosa* was 100, 75, 250, and 200 μg/mL, respectively, compared to 150, 100, 300, and 250 μg/mL for curcumin. The authors point out that the greater activity of nanocurcumin compared to curcumin in DMSO is related to particle size. Curcumin in nanoparticle form has a reduced size of 2–40 nm, much smaller than curcumin particles dissolved in DMSO (500–800 nm). This leads to better penetration and greater absorption by bacteria [21]. In recent times, antibacterial photodynamic therapy has emerged. Agel et al. (2019) have been exploring the efficacy of curcumin-loaded PLGA nanoparticles to enhance and increase its activity against bacteria. A reduction in CFU was observed for curcumin-loaded nanoparticles and chitosan-modified curcumin-loaded nanoparticles. Furthermore, by changing the surface charge of the nanoparticles using chitosan, the authors obtained an increased photodynamic efficacy against *E. coli* from 0.13% bacterial survival for uncoated nanoparticles to 0.00013% bacterial survival for chitosan-modified nanoparticles [22].

Additionally, the development of nanovectors capable of encasing oxygen can be crucial for boosting the generation of ROS during PDI [22]. In fact, the PDI therapeutic approach consists of the illumination of a photosensitive substance with light at a specific wavelength, leading to the production of ROS, which is able to inactivate various bacteria, reducing the possibility for pathogens to develop resistance mechanisms [23]. 

Moreover, oxygen delivery can play a key role in wound healing processes. Previously, oxygen-loaded chitosan shelled nanodroplets purposely developed for the treatment of infected chronic wounds showed a good biocompatibility with human keratinocytes, alongside prolonged oxygen delivery [24]. Prato et al. showed that oxygen-loaded nanobubbles can increase the skin tissue oxygen level. The authors described nanobubbles capable of storing oxygen, which appears as a possible innovative, economical, and non-toxic therapeutic tool for hypoxia-associated skin pathologies [25]. 

Given this background, in the present study, we successfully manufactured oxygen-loaded chitosan-shelled nanobubbles with an optimal curcumin encapsulation capability either dissolved in the core or by chemical conjugation to the shell. The NBs can protect curcumin from degradation and can deliver the compound with sustained release kinetics. In addition, the natural fluorescence of curcumin allowed us to obtain fluorescent labelled formulations suitable for fluorescent microscopy studies. 

Microbiological tests were conducted to assess the antibacterial effects of NB formulations on the three investigated strains, i.e., Gram-positive *S. aureus* and *Enterococcus faecalis* and Gram-negative *E. coli*, considering short (minimal inhibitory concentration, MIC, and minimal bactericidal concentration, MBC) and long time (time kill) effects due to simple administration and following LED photoactivation.

To further investigate the different responses from the selected strains, we assessed the mutual interaction of the bacteria and the NBs by confocal imaging to evaluate whether some internalization occurred and to what extent the bacterial membranes were trespassed, causing local damages, as independently documented by the lactate dehydrogenase activity (LDH) test, which quantifies a cell’s membrane leakage/opening with the release of lactate dehydrogenase in the culture medium. 

Finally, to better understand which mechanism related to NBs is mainly responsible for bacteria membrane damage, both the ROS production and the lipid oxidation capability were evaluated following the LED photoactivation.

## 2. Results

Dual-loaded chitosan-shelled NBs have been prepared, containing oxygen in the core and curcumin either in the core or in the shell. Chitosan was selected as an NB shell for it intrinsic antimicrobial effects and for the possibility of subsequent polymer functionalization. The conjugation of curcumin to chitosan was carried out by exploiting the formation of a Schiff base.

### 2.1. Physicochemical Characterization of NBs Formulations

Dynamic light scattering (DLS) was used to determine the physicochemical parameters of the examined NBs, including average diameter, polydispersity index, and zeta potential. As shown in Table 1, NBs had an average size of about 500 nm and their diameter somewhat increased after the oxygen was added.

The presence of chitosan on the NBs’ surface is responsible for their positive zeta potential values. Curcumin addition reduced the zeta potential in comparison to the one with the blank NB formulation. The values, however, continued to be high enough to avoid NB aggregation. All of the NBs’ pH levels, which ranged from 4.8 to 5.4, were evaluated as well. The physical stability of NB formulations was investigated over time after storage at 4 °C in the dark.

The overall diameter of all formulations exhibited a small rise after a 2-month testing period (Figure 1). The sizes expanded by 9.20% to 12.19%, on average. Even after 2 months, the diameter of the NBs remained below 600 nm, despite the statistical significance of the size shift. The pH and zeta potential values were found to be equivalent to the values in Table 1, even after 2 months.

### 2.2. Morphological Studies

All of the NB formulations underwent morphological analysis using fluorescence microscopy. The chitosan-shelled, curcumin-loaded-NBs with conjugated curcumin showed a spherical shape with clear and well-defined shells, as shown in Figure 2. A well-defined fluorescent shell was observed in the NB nanostructure, confirming the presence of curcumin-conjugated chitosan on the NB surface.

Moreover, the fluorescence microscopy image confirmed the NB size, measured by DLS analysis.

### 2.3. Encapsulation Efficiency and Loading Capacity Determination

All prepared NBs underwent high-performance liquid chromatography (HPLC) analysis to measure the curcumin concentration. The encapsulation efficiency (EE) and loading capacity (LC) of NB formulations are shown in Table 2. 

Curcumin was successfully encapsulated in the core and within the chitosan shell. The LC values were under 10% for all formulations. All NB EE ranges were between 80% and 88%. Both EE and LC displayed a small increase when curcumin was also present in the shell. Compared to Curcumin-NBs, Curcumin-Oxygen-NBs showed a slight rise in both EE and LC in the chitosan shell. As a result, oxygen had no influence on the loading capacity or encapsulation efficiency of any of the formulations. It is important to note that a 0.1% relative error was predicted based on the accuracy of the equipment.

### 2.4. In Vitro Release of Curcumin

Studies on the in vitro release of curcumin from Curc-CS-NBs and Curc-Oxy-CS-NBs were conducted. The findings of the release research are reported in Figure 3. For both formulations, the percentage of curcumin released during a prolonged time period was assessed. The amount of curcumin released from Curc-CS-NB and Curc-Oxy-CS-NB after 48 h was 69.14% and 66.98%, respectively. These findings show that the curcumin is well incorporated and released from both formulations with sustained kinetics, which suggests prolonged activity over time.

### 2.5. Microbiological Study 

#### 2.5.1. Minimum Inhibitory Concentration (MIC) and Minimum Bactericidal Concentration (MBC)

The antibacterial activity of Curc-CS-NBs and Curc-Oxy-CS-NBs was examined, and the MIC and MBC were evaluated with and without photodynamic activation using LED light (Table 3). The results demonstrated that, after photoactivation, both types of NBs exhibited increased effectiveness against all tested bacterial strains, particularly *S. aureus* and *E. faecalis*.

In the instance of *E. coli*, Curc-CS-NBs had a MIC range of 46 µg/mL, whereas Curc-Oxy-CS-NBs displayed a MIC range of 93 µg/mL in the absence of photodynamic activation. However, when both kinds of NBs were exposed to LED light, the MIC for both formulations dropped to 46 µg/mL.

The MIC values of Curc-CS-NBs and Curc-Oxy-CS-NBs for *S. aureus* were 93 µg/mL and 46 µg/mL, respectively, without photodynamic activation. The MIC for Curc-CS-NBs remained at 46 µg/mL following photodynamic activation, whereas it drastically dropped to 12 µg/mL for Curc-Oxy-CS-NBs.

The MIC values for *E. faecalis* without photodynamic activation were 46 and 93 µg/mL for Curc-CS-NBs and Curc-Oxy-CS-NBs, respectively. Following LED illumination, the MIC values for Curc-CS-NBs and Curc-Oxy-CS-NBs decreased to 23 and 46µg/mL, respectively.

#### 2.5.2. Time Kill Kinetics

At concentrations three times the MIC of curcumin, Curc- CS-NBs and Curc-Oxy-CS-NBs were tested for their antibacterial efficacy against the three bacterial strains. The studies were carried out following three hours in the dark or under LED illumination at various time intervals (T0, 90 min, 6 h, 20 h, 24 h, 48 h, and 72 h). By contrasting the bacterial samples that had been exposed to NBs with the control (untreated) samples, the percentage of survival was determined.

According to the findings, all bacterial strains were more effectively killed by NBs that had both oxygen and LED irradiation. Furthermore, the MIC value had an impact on the killing potential: the greater the MIC values, the higher the killing activity.

##### *Escherichia coli* ATCC 25922

The findings of the *E. coli* time kill kinetics demonstrated the significance of a number of factors, including the MIC, for successfully limiting the bacterium’s spread over a prolonged period. The red color in Figure 4B shows that Curc-Oxy-CS-NBs at a low MIC concentration (46.4 µg/mL), in combination with 3 h of LED irradiation, were capable of continually killing bacteria for several days. An interesting fact about this MIC concentration was that it was three times lower than the MIC concentration of curcumin alone [6]. Therefore, our results suggest that the usage of curcumin-loaded NBs in conjunction with oxygen and LED light can efficiently eliminate even Gram-negative bacteria such as *E. coli.*

It should be illustrated that the antibacterial activity of the NBs was not prominently apparent when evaluating the MIC and MBC alone, since these measurements were basically instantaneous and the release of curcumin from the NBs was minimal (Figure 4). However, in the time killing test, the prolonged release of curcumin (as well as oxygen) from the NBs, along with their activation by light, enhanced the antibacterial action over a longer period of time.

##### *Staphylococcus aureus* ATCC 29213 

Overall, the results indicate that curcumin-loaded NB formulations, both with and without oxygen, are very efficient in reducing *S. aureus* growth. An MIC value of 12 µg/mL was adequate to limit bacterial growth for a prolonged period of time when Curc-Oxy-CS-NBs were subjected to LED irradiation (Figure 5B). However, a higher MIC value of 93 µg/mL was required to establish equivalent control over bacterial growth for a duration of 3 days in the absence of LED illumination (Figure 5A).

##### *Enterococcus faecalis* ATCC 29212 

Results with *E. faecalis* further showed the Curc-Oxy-CS-NBs’ improved killing ability at the MIC value of 46 µg/mL (Figure 6B). These results suggest that the efficient inhibition of bacterial growth requires the interaction of curcumin, oxygen, and LED irradiation. The results found that the NBs were unable to sustain the bactericidal activity for a long time if any one of these specific factors was missing.

### 2.6. Confocal Microscopy Study 

To investigate the different membrane damage induced by the NBs following irradiation, and to detect whether the NBs are effectively internalized or whether they cover the external membrane of the bacteria, we performed a detailed analysis of the geometry of their interaction by confocal microscopy.

Confocal microscopy (Figure 7) showed that, starting from the first observation (3 h), and also, subsequently, (24 h), the Curc-CS-NBs seemed to adhere to the wall of *E. coli* without being internalized (A), while the Curc-CS-NBs appeared to be eagerly taken up and internalized by staphylococci (B) and enterococci (C). These findings provide valuable insights into the specific interactions of Curc-CS-NBs with different bacterial strains, offering potential approaches for enhancing antimicrobial efficacy and targeting specific pathogens.

### 2.7. Lactate Dehydrogenase Activity

The activity of lactate dehydrogenase (LDH) was examined as an indicator of bacterial membrane integrity to confirm the results of the time kill investigations. The findings showed that Cur-Oxy-CS-NBs severely compromised bacterial membrane integrity when subjected to LED irradiation, causing the release of the cytosolic enzyme LDH. Figure 8A–C show these findings.

For Curc-CS-NBs and Curc-Oxy-CS-NBs, the proportion of LDH released in a dark environment ranged from 6% to 49%. However, the percentage of LDH released for Curc-NBs and Curc-Oxy-CS-NBs after LED application varied from 10% to 85%. The three bacterial strains that were examined had different membrane damage onset times. The observed impacts on bacterial membranes and the subsequent antimicrobial activity can be explained by the presence of oxygen and by LED irradiation, which causes the production of reactive oxygen species (ROS).

To better understand the mechanism of damage, we investigated whether the NBs were responsible for lipid peroxidation of the membrane and evaluated the entity of ROS production at different curcumin concentrations.

### 2.8. Antioxidant Activity Evaluation: Malondialdehyde Test

The antioxidant capability of curcumin loaded in NBs was investigated. The results are reported in Figure 9. The formation of malondialdehyde (MDA), as a marker of oxidative stress, was dramatically decreased when curcumin-loaded nanobubbles were used in comparison to blank NBs (*p* < 0.001). This marked decrease in MDA production demonstrated that the antioxidant activity of curcumin is maintained, even in NBs. This behavior raises the possibility that curcumin inhibits tissue oxidation.

### 2.9. ROS Production 

The objective of the current research is to ascertain how the presence of curcumin-containing NBs, both with and without oxygen, as well as their exposure to LED irradiation, impact the generation of ROS, which play crucial roles in cellular processes. In fact, an excessive amount of ROS production can cause oxidative stress, which can damage cells. 

Various formulations or conditions were tested, with different concentrations of curcumin diluted in the same brain heart infusion (BHI) media used in the microbiological study and at comparable concentrations. The total amount of fluorescence emitted by the sample (mean fluorescence intensity), which is expressed in RFU, is related to the amount of ROS. The data also includes information about irradiation duration (1 or 3 h). Overall, the data refers to the evaluation of the effects of various curcumin formulations and timings of irradiation on the level of ROS.

The study’s results show that when the curcumin formulations were subjected to 3 h of LED irradiation and when oxygen was present in the NBs, ROS generation was greater. The study’s findings also showed that ROS production was correlated with a drop in curcumin content in NBs (Figure 10).

## 3. Discussion

The main goal of this work is to develop an innovative dual approach for dealing with the serious problem of bacterial infection control. The strategy involves the use of natural compounds to combat antibiotic resistance, which has emerged as a critical concern for public health.

In this context, two innovative steps were explored in the present study. At first, curcumin was selected as a natural compound because we previously evaluated its efficient photoactivation following LED irradiation using a curcumin cyclodextrin solution [6].

In addition, its antibacterial activity in nanoliposomes associated with photodynamic therapy was recently demonstrated [26]. 

Besides, curcumin presents some limitations to take into account for the formulation design, such as poor water solubility, stability, and low bioavailability. Much research investigated the incorporation of curcumin into biocompatible nanosystems, such as polymeric nanoparticles and nanoemulsions, to overcome these problems [22,27,28,29].

Here, curcumin was administered using purposely tuned chitosan-shelled NBs as nanovectors for resolving difficulties with curcumin’s solubility and chemical degradation. The curcumin NB incorporation was improved, exploiting two different NB domains. Interestingly, a significant amount of curcumin can be incorporated when it is loaded both conjugated to the polymer shell and within the core of the NBs [27]. 

Preliminary data obtained by our group showed the possible synergistic effect of curcumin combined with photodynamic treatment for 3 h by MIC at a very low concentration of curcumin, i.e., 0.125, <0.0075, and <0.0037 mg/mL against *E.coli*, *S. aureus*, and *E. faecalis*, respectively [6]. Wang et al. report an overview of the effectiveness of antimicrobial blue light inactivation of different microbes, its mechanism of action, and the potential development of resistance to blue light by microbes [23].

Studies conducted by Bhawana et al. demonstrate that the aqueous dispersion of nanocurcumin was much more effective than curcumin against *S. aureus*, *E. coli*, *Pseudomonas aeruginosa*, *Penicillium notatum*, and *Aspergillus niger* [21]. 

The developed nanoformulation has the capability to store oxygen in the decafluoro pentane core, as previously reported [30]. Curcumin-loaded chitosan-shelled NBs were then loaded with oxygen to evaluate if a synergic antimicrobial effect might be achieved by combining the two molecules. The co-loaded chitosan-shelled NBs represent a dual nanodelivery system able to produce increased benefits by exploiting the two combined mechanisms. Moreover, the chitosan shell is able to improve adhesiveness to bacterial membranes by utilizing the well-known mucoadhesive capability of the nanobubbles [31]. 

The chitosan-shelled NB adhesion can favor the localized and continuous release of curcumin, which is accomplished through sustained release kinetics. The mechanism behind oxygen release from NBs involves the passive diffusion of gas via the shell, primarily influenced by the variance in gas concentration between the core and the external environment. It is noteworthy that, in Curc-Oxy-CS-NBs, the core, which is made up of decaperfluoropentane with high oxygen solubility capability, serves as a reservoir from which oxygen is progressively slowly released, with its kinetics based on partial pressure and according to Henry’s law. Furthermore, the release kinetics are influenced by the substantial internal NB pressure, which is inversely proportional to the bubble radius, as per the Laplace law [30,32].

Following the successful demonstration of satisfactory stability and the maintenance of curcumin’s antioxidant properties, the antibacterial efficacy of two nanoformulations, namely Curc-CS-NBs and Curc-Oxy-CS-NBs, was evaluated.

Taking into account the microbiological studies, it is important to keep in mind that the various bacterial strains under examination may have distinct particular characteristics and structures [33]. As a consequence, three different types were investigated.

Upon conducting a comparison of their MIC values with those documented in [6], it becomes evident that the MIC values of the nanoformulations were similar to those observed with curcumin alone for *S. aureus*. This indicates that, even though *S. aureus* already demonstrated high responsiveness to curcumin, the nanoformulations are equally effective in inhibiting the growth of this bacterium. On the contrary, for *E. faecalis* and *E. coli*, the MIC values of the nanoformulations are significantly lower than those of bulk curcumin.

The results of the study indicate that when comparing bulk curcumin to nanoformulations, the latter exhibits significantly enhanced absorption and penetration of the curcumin. The curcumin nanoformulations show a much higher ability to permeate and be absorbed by the target tissues or cells, making it more effective in delivering the curcumin compound to its intended site of action.

A comparison of the time kill kinetics is necessary to assess the effects of slow release, since both the bulk and nanoencapsulated curcumin were utilized at dosages that were three times the MIC.

In the case of *S. aureus*, Curc-CS-NBs and Curc-Oxy-CS-NBs both shown great effectiveness over the entire period of the study, thereby enhancing the already outstanding outcomes seen with the administration of bulk curcumin.

However, the 72-h control of *E. faecalis* and *E. coli* growth was less successful. This might be explained by the fact that NBs were less effective against them, due to the lower MIC values and total curcumin dosage supplied, as compared to bulk curcumin. 

This study’s second major breakthrough is the association of NB formulation with PDI photoactivating curcumin under blue LED light. The objective of this strategy is to take advantage of the bactericidal properties that ROS generation provides. 

As demonstrated by the observed rise in bacterial membrane leakage, as shown by the LDH test, the use of Curc-Oxy-CS-NBs increased the treatment’s toxicity.

According to the microbiological findings, Curc-CS-NBs and Curc-Oxy-CS-NBs had MIC values that were greater than those predicted for the ‘bulk’ curcumin in the two Gram-positive bacteria after 3 h of LED illumination. However, the MIC values for the nanoformulations in the case of *E. coli* were considerably lower.

In light of the fact that the MIC value was determined within 3 h after the administration of the NBs, the results for *E. faecalis* and *S. aureus* are in line with expectations, since the amount of curcumin that was released and accessible for photoactivation would be considered somewhat low at this early stage.

The time kill test findings meet our expectations for the bactericidal activity for all bacterial strains over a 72-h period, especially for Curc-Oxy-CS-NBs. The outcomes show a considerable bactericidal impact, notably for *E. coli*, despite the lower dose of curcumin. These results indicate the efficacy of the treatment, as the results are compatible with the measured LDH release and generation of ROS.

The powerful photodynamic reaction of curcumin to light has been demonstrated In earlier studies. Researchers have shown that curcumin does not have any noticeable antibacterial properties when used alone or in the absence of light. Agel et al. [22] generated curcumin-loaded nanoparticles, as well as curcumin-loaded nanoparticles integrating chitosan, in order to assess the antibacterial photodynamic effects of curcumin on particular bacterial strains such as *Staphylococcus saprophyticus subsp. Bovis* and *E. coli* DH5 alpha. The in vitro experiments were supported by the increased adherence of chitosan-modified nanoparticles to bacteria and the observed structural damage resulting from photodynamic activation. Moreover, the survival rate of *S. saprophyticus* was significantly reduced to less than 0.0001% (corresponding to a >6.2 log10 decrease) when curcumin-loaded nanoparticles were utilized in combination with LED irradiation. Notably, the effectiveness was further enhanced when LED light and nanoparticles containing curcumin and chitosan were combined, resulting in a survival rate of 0.0000045%. In contrast, when tested on *E. coli*, the photoactivated curcumin-loaded nanoparticles exhibited a survival rate of 0.13% (corresponding to a 2.9 log10 reduction in CFU). However, an enhanced antibacterial efficacy was observed when utilizing curcumin-loaded nanoparticles with chitosan in combination with LED light, resulting in a remarkable reduction in CFU/mL up to 5.9 log10 (corresponding to a survival rate of 0.00013%) [22]. 

Interestingly, the findings of this study demonstrate the remarkable capability of Curc-Oxy-CS-NBs (NBs loaded with curcumin and oxygen and coated with chitosan) to effectively damage Gram-negative bacteria, such as *E. coli*. Gram-negative bacteria possess a unique cell envelope structure, including an outer membrane with lipopolysaccharides, which acts as a permeability barrier and provides resistance against antimicrobial agents [34,35,36]. However, the application of Curc-Oxy-CS-NBs combined with blue LED light overcame these challenges. The chitosan shell of the nanovector facilitated the penetration of curcumin, enabling its diffusion and photosensitization within the bacterial cells. This novel approach holds promise for targeting and combating Gram-negative bacteria that are typically difficult to eradicate [22,34,37].

Another result of the current research provides valuable insights into the impact of curcumin-containing nanoformulations, both with and without oxygen, and their exposure to LED irradiation on the generation of ROS. It was observed that an increase in ROS generation when curcumin-containing nanoformulations were exposed to 3 h of LED irradiation in the presence of oxygen suggests that these conditions promote a more pronounced pro-oxidant effect. This is consistent with previous studies which show that prolonged LED exposure and the presence of oxygen can lead to heightened ROS production due to increased photoactivation and oxygen singlet formation. The synergistic effect observed in this study emphasizes the need to carefully consider the duration of LED irradiation and oxygen levels when developing curcumin-based therapies, as elevated ROS levels could potentially contribute to oxidative stress and cellular damage. Additionally, the correlation between ROS production and decreased curcumin content within the nanoformulations raises concerns about curcumin stability under certain conditions. Future research should focus on elucidating the underlying mechanisms of curcumin degradation or transformation in nanostructures to ensure the formulation’s efficacy and bioavailability over time. These findings are crucial for the development of safer and more effective curcumin-based treatments that harness its therapeutic potential while mitigating any potential oxidative harm. Overall, the study contributes valuable insights into ROS regulation in curcumin-containing nanoformulations and offers a foundation for further investigations to optimize their biomedical applications.

Overall, our results show that these chitosan-shelled nanobubbles have a lot of potential as a platform for targeted curcumin delivery, showing a prolonged release and increased antimicrobial properties through the combination of light activation and ROS formation.

In conclusion, the aim of this research was to optimize chitosan-shelled nanobubbles for the co-loading of curcumin and oxygen, and to evaluate their antimicrobial activity. The newly developed Curc-CS-NBs and Curc-Oxy-CS-NBs exhibited favorable physicochemical properties and demonstrated strong antimicrobial effects, which were further enhanced through photodynamic activation using visible light, specifically LED. The combination of Curc-Oxy-CS-NBs and photodynamic treatment resulted in a more effective inactivation of the microorganisms compared to conditions without oxygen and LED exposure.

Hence, the overall findings of this study hold significant implications in combating pathogenic microorganisms that affect the skin and other external mucosal surfaces. The demonstrated efficacy of photoactivated curcumin and oxygen-loaded NBs in eradicating common and widespread bacterial strains highlights their potential in defending against such pathogens. Furthermore, there is a need for further investigation into the application of nanobubbles for food preservation, targeting both bacteria and fungi. Future research could focus on optimizing the experimental conditions by reducing the curcumin content in the NBs while maintaining their effectiveness and exploring shorter photodynamic activation periods that still yield potent antibacterial activity. These findings contribute to a broader strategy aimed at addressing pathogenic microorganisms, safeguarding human health, and enhancing global food quality.

## 4. Materials and Methods

### 4.1. Chemicals

Decaperfluoropentane and ethanol were obtained from Sigma-Aldrich (St. Louis, MO, USA). Epikuron 200^®^ was kindly provided by Cargill Texturizing solutions, Germany. Palmitic acid was obtained from Fluka (Buchs, CH, Switzerland). Curcumin (Molecular weight = 368.4 Da), chitosan, and 1-methyl-2-pyrrolidinone were sourced from various suppliers. Trypticase soy broth and agar (TSB, TSA) and Sabouraud dextrose (SAB) broth and agar were provided by Oxoid SpA (Garbagnate Milanese, Italy). 

The bacterial strains used in this study were obtained from the American Type Culture Collection (ATCC). The strains included *E. coli* ATCC 25922, *S. aureus* ATCC 29213, and *E. faecalis* ATCC 29212.

The selected strains were cultured on specific agar media provided by Oxoid SpA: mannitol salt agar (MSA) for *S. aureus*, brain heart infusion agar (BHA) for *E. faecalis*, and MacConkey agar (MAC) for *E. coli*. The agar plates were then incubated at 37 °C for 24 h. Young colonies were collected and adjusted to approximately 3–4 McFarland standard. The bacterial suspensions were transferred to cryovials containing a cryopreservative fluid and porous beads to facilitate bacterial adherence. The cryovials were stored at −80 °C for long-term storage using Micro-bank (PRO-LAB Diagnostic, Richmond Hill, ON, Canada, system.).

The LDH release assay was carried out using the LDH cytotoxicity test kit (Cayman Chemical # 601170). All other chemicals used in this study but not listed above were of analytical grade and were commercially available.

### 4.2. Instrumentations

#### LED System

In this study, a blue-light-emitting diode (LED) with a wavelength range of 425 nm to 470 nm was utilized. The specific LED model used was the LES Flex Strips LEDYDEL IP64, sourced from Turin, Italy (Figure 11).

### 4.3. Preparation of Curcumin Conjugated Chitosan

The curcumin-conjugated chitosan (CCC) was prepared using the imine formation (Schiff base reaction) method [38,39,40,41,42]. To form the conjugate, chitosan (2.7% *w*/*w*) was dissolved in acetic acid buffer at pH 4.5 while stirring at 500 rpm. The chitosan mixture was then combined with a 0.1% curcumin solution dissolved in 5 mL of ethanol. The reaction mixture was continuously stirred at 700 rpm for 30 min, and then the temperature controller was set to 50 °C. Stirring and heating at 60 °C were maintained overnight. Finally, the reaction mixture was cooled to room temperature. This CCC was used as a shell in the NBs.

### 4.4. Preparation of NB Formulations

Curcumin-loaded chitosan-shelled NBs, both with and without the presence of oxygen in the core, were prepared. Blank chitosan-shelled NB were also formulated for comparison purposes. Table 4 reports the three types of nanobubbles produced.

To obtain the blank NB formulation, a mixture was prepared by dissolving Epikuron 200^®^ (Cargill) (3% *w*/*w*) and palmitic acid (0.5% *w*/*w*) in ethanol. After this, 210 µL of decaperfluoropentane and an appropriate amount of distilled water were added to the mixture and the sample was homogenized using an Ultra-Turrax^®^ high-strength homogenizer for 2 min, resulting in the formation of a nanoemulsion. Subsequently, a drop-by-drop addition of an aqueous solution of chitosan (2.7% *w*/*w*, pH 4.5) was carried out to form the polymeric shell of the blank NBs. 

The curcumin NBs were obtained by loading curcumin both in the core and in the shell. For this purpose, a curcumin solution in N-methyl-2-pyrrolidone was added to the Epikuron 200^®^ and palmitic acid mixture and curcumin-conjugated chitosan (CCC) was added to obtain the NB shell. For the preparation of oxygen-loaded curcumin NBs (Curc-Oxy-CS-NBs), the formulation was saturated with O_2_ for 10 min before the polymer shell deposition. Then, the curcumin conjugated chitosan was added dropwise to the sample kept in magnetic stirring and under oxygen purge.

### 4.5. In Vitro Characterization of NB Formulations

The average size, polydispersity index, and zeta potential of the nanoformulations were determined using photon correlation spectroscopy (PCS) and dynamic light scattering (DLS) with a 90 Plus particle sizer (Brookhaven Instruments Corporation, Holtsville, NY, USA) equipped with Mas option particle sizing software. The measurements were performed at a temperature of 25 °C and a fixed scattering angle of 90°.

Fluorescence microscopy was employed to analyze the morphology of each formulation of NBs. This technique leveraged the intrinsic fluorescence properties of curcumin, which emits light with a peak wavelength of approximately 543 nm [43]. By utilizing fluorescence microscopy, the structural characteristics and arrangement of the NBs could be visualized and examined in detail. 

#### 4.5.1. Determination of Curcumin Concentration with HPLC

For the quantitative determination of curcumin and the release of curcumin from nanobubbles, high-performance liquid chromatography (HPLC) was conducted using a Shimadzu system equipped with a UV/Vis detector. The HPLC system utilized a reverse phase Agilent RP-C 18 column (250 cm × 4.6 mm, pore size 5 μm). The mobile phase consisted of acetonitrile and water in a ratio of 70:30 *v*/*v*. A flow rate of 1 mL/min was maintained to propel the mobile phase through the column. The detection of the curcumin peak was performed at a wavelength of 425 nm. Curcumin peaks were observed at a retention time of approximately 13 min. 

To calculate the concentration of curcumin in the nanobubbles, standard calibration curves were utilized. Firstly, a specific amount of standard curcumin was weighed and mixed with methanol in a volumetric flask to create a stock standard solution. This solution was then diluted with the mobile phase to generate a series of standard solutions up to a concentration of 50 μg/mL, which were subsequently injected into the HPLC system. The resulting data yielded linear calibration curves for curcumin with regression coefficients of 0.999.

To determine the curcumin concentration in the nanobubbles, a chloroform extraction procedure was employed. This involved combining 3 mL of the appropriate nanobubbles with 3 mL of chloroform, followed by vortexing the mixture for 1 min and cooling it in an ice bath. Once the extraction was complete, the chloroform was evaporated using a nitrogen pump. The dried curcumin was then dissolved in 1 mL of methanol and analyzed by HPLC.

#### 4.5.2. Encapsulation Efficiency and Loading Capacity Determination

To determine the encapsulation efficiency of curcumin-loaded nanobubbles (NBs), a centrifugal filter device (Amicon Ultra-0.5 centrifugal filter device, Merck KGaA, Darmstadt, Germany) was used. After centrifugation of 200 μL of the NBs at 15,000 rpm for 30 min using a Beckman Coulter Allegra 64R Centrifuge, the concentration of free curcumin in the ultrafiltrate was determined via HPLC analysis. Encapsulation efficiency was determined by subtracting the amount of free curcumin from the initial added amount using Equation (1):% Encapsulation efficiency (EE) = (Total drug − Free drug)/(Total drug) × 100(1)

To determine the loading capacity of the nanobubbles (NBs), freeze-dried NB samples weighing approximately 5 mg were utilized. These samples were mixed with 1 mL of ethanol/methanol and subjected to 15 min of sonication. After sonication, the samples were centrifuged at 15,000 rpm for 5 min. The absorbance of the supernatant was measured at 425 nm using a spectrophotometer, and Equation (2) was applied to calculate the loading capacity. Additionally, the loading capacity was verified using high-performance liquid chromatography (HPLC) with the extraction method.
% Loading Capacity (LC) = (amount of drug in NB)/(freeze − dried NB weight) × 100 (2)

#### 4.5.3. In Vitro Curcumin Release Study

To assess the in vitro release of curcumin from curcumin-loaded nanobubbles (NBs), a multi-compartment rotating cell system was employed. The curcumin-loaded NBs were placed inside the donor chambers of the system. To separate the donor and receiving chambers, a semi-permeable cellulose membrane with a cut-off of 14 KDa was utilized. In the experiment, the receiving phase consisted of a 5% *w*/*v* solution of beta-cyclodextrin for all formulations.

During the 48-h experiment, 1 mL samples were periodically extracted from the receiving phase at predetermined intervals. After each extraction, the receiving phase was replenished with 1 mL of the beta-cyclodextrin solution. The release of curcumin was monitored using HPLC with a Shimadzu UV/VIS detector (SPD-20AV, Markham, ON, Canada).

This setup allowed for the evaluation of the release kinetics of curcumin from the nanobubbles over time, providing insights into the sustained release behavior of the formulations.

### 4.6. Microbiological Assay

The minimum inhibitory concentration (MIC) and minimum bactericidal concentration (MBC) of NBs, with and without exposure to 3 h of LED photodynamic activation, were determined using the microdilution method as per the CLSI guideline (CLSI, 2007). Overnight cultures of each bacterial strain were prepared, and the inoculum was adjusted to a turbidity of 0.5 McFarland standard (equivalent to 5 × 10^8^ CFU/mL). This inoculum was further suspended in BHI media to obtain a concentration of 10^5^ CFU/mL for use in the microdilution plates containing diluted test compounds (curcumin, NBs).

Two microtiter 96-well plates were used for the experiment. The first well of the first plate contained an appropriate concentration of the test compound, which was then serially diluted across the plate. Each well of the microtiter plate contained different concentrations of the respective test compound. One plate was kept in the dark, while the other plate was exposed to an LED system for three hours. After incubation for 24 h at 37 °C, a 10 µL aliquot was taken from each well that showed no visible growth after incubation. These aliquots were spotted onto BHA and incubated at 37 °C for 24 h. The MBC was determined as the lowest concentration of nanobubbles that completely inhibited bacterial growth on the agar plates.

#### 4.6.1. Time Kill Kinetics

The bactericidal or bacteriostatic activity of antimicrobial nanobubbles against a specific bacterial strain was evaluated using the Time-kill Kinetics test, as described in reference [44]. A bacterial inoculum with a concentration of 10^5^ cells/mL was prepared from an overnight culture for each bacterial strain. Six-well microtiter plates were prepared using BHI media, with each well containing the respective test compound at a concentration of 3 times the MIC, along with the bacterial inoculum. After one hour of incubation in a hood, one plate was exposed to LED irradiation, while the other plate was kept in the dark. The time kill kinetics were assessed at different time points, including T0 (initial time), 6 h, 20 h, 24 h, 48 h, and 72 h.

#### 4.6.2. Evaluation of Nanoformulation Uptake by Bacteria through Confocal Microscopy Analysis

Bacteria were cultured in brain heart infusion (BHI, Oxoid, Italy) broth at 37 °C to a concentration of 1 × 10^9^ colony forming units (CFU)/mL. Then, 1 mL aliquot of *E. coli*, *S. aureus*, and *E. faecalis* was pelleted (3000 g × 10 min at 4 °C), resuspended in PBS 1×, and incubated without (control) or with 10% *v*/*v* 6-coumarin-labeled NPs (cNPs) for 3 h and 24 h with agitation (160 rpm) in the dark at 37 °C. 

After incubation, controls and bacteria cNPs suspensions (50 μL) were transferred on poly-l-lysine-coated microscope slides and allowed to dry. Then, bacteria were stained with 5 μg/mL iodide propidium (PI, Invitrogen) in PBS 1× and again allowed to dry in a humid chamber at 37 °C for 15 min.

For microscopy analysis, bacteria were mounted in a mounting medium and covered by cover slips. Imaging was performed using a Leica TCS SP8 confocal system (Leica Microsystems) equipped with an argon ion and 561 nm DPSS lasers. Images were acquired using a HCX PL APO 63×/1.4 NA oil immersion objective with a resolution of 0.07 μm × 0.07 μm.

#### 4.6.3. Cell Membrane Damage, ROS Production, and Oxidative Stress Evaluation 

##### Lactate Dehydrogenase Release Assay 

Lactate Dehydrogenase (LDH) is an enzyme that remains stable within cells but is rapidly released into the culture medium when the plasma membrane is disrupted or damaged [44]. The LDH test, commonly referred to as the LDH release assay, is a cytotoxicity/cell death assay that measures the extent of plasma membrane damage.

To detect LDH leakage into the cell culture medium, an assay incorporating a tetrazolium salt is employed. This assay involves two sequential reactions. In the first step, LDH facilitates the conversion of NAD+ to NADH and H+ by oxidizing lactate to pyruvate. In the second step, the newly formed NADH and H+ react with an electron acceptor in the presence of a tetrazolium salt (INT), resulting in the reduction of INT to a brightly colored formazan compound that exhibits significant absorbance at 490–520 nm [44]. The quantity of formazan produced is directly proportional to the amount of LDH released, serving as a marker of cytotoxicity.TInizio modulo

To conduct the cytotoxicity assay, an overnight culture was used to prepare a bacterial inoculum with a concentration of 10^5^ cells/mL for each bacterial strain. Two microtiter plates, each containing twenty-four wells, were prepared by combining BHI media, the respective test compounds (NBs, Curcumin) at a concentration of 3MIC, and the bacterial inoculum. After one hour of incubation in a hood, one plate was exposed to LED light for three hours, while the other plate was kept in the dark at room temperature. Following the three-hour exposure, the cytotoxicity protocol was initiated.

All samples, including spontaneous release and maximum release controls, were centrifuged for 10 min at 13,000 rpm. The spontaneous release control was obtained by mixing 200 µL of bacterial suspension with 20 µL of assay buffer, while the maximum release control was prepared by mixing 150 µL of bacterial suspension with 45 µL of Triton X-100 (3%). The resulting supernatant (100 µL) was transferred to a new 96-well assay plate, and 100 µL of LDH Reaction Solution was added to each well. The plate was then incubated at 37 °C for 30 min with gentle shaking on an orbital shaker. After the incubation period, the optical density (O.D) was measured at 490 nm using a microplate reader. LDH activity was assessed at T0, 4 h, and 24 h after incubation. The results of each experiment were expressed as “% cytotoxicity”, which represented the proportion of LDH present in the target cells. Spectrophotometry was employed for data evaluation using Equation (3):% Cytotoxicity = [(Experimental Value) − (Spontaneous Release)/(Maximum Release) − (Spontaneous Release)] × 100(3)

##### Determination of Reactive Oxygen Species (ROS)

For ROS quantification, a fluorometric method, based on the oxidation of the fluorescent probe 2’,7’-dichlorofluorescin, was used. For the 2’,7’-dichlorofluorescin test, a 400 µM H2DCF solution was prepared by mixing 0.5 mL of a 10 mM 2’,7’-Dichlorodihydrofluorescein diacetate (H2DCFDA) ethanolic stock solution with 2 mL of NaOH 0.01 M. The hydrolysis product, H2DCF, was kept at room temperature for 30 min and neutralized with 10 mL of 50 mM phosphate buffer (pH 7.2). This solution was freshly prepared and kept on ice prior to use. In the presence of ROS, H2DCF is rapidly oxidized in the fluorescent DCF (2’,7’-dichlorofluorescin). The fluorescence intensity of all standards and samples was measured using a spectrofluorometer (EnSightTM automated multimode plate reader, Perkin Elmer) at the emission and excitation wavelengths of 485 and 530 nm. The concentration of ROS was calculated by using a calibration curve obtained by analyzing DCF standard solutions in the concentration range between 0.001–1 µM. For ROS determination in the samples, a volume of H2DCF solution (400 µM) was added to each sample to obtain final concentrations of 5 µM. Samples were allowed to equilibrate at room temperature in the dark for at least 20 min to allow the reaction to be completed. The intensity of fluorescence was measured before 60 min. All the experiments were performed in triplicate.

##### Malondialdehyde Test (Oxidative Stress Evaluation)

The assessment of antioxidant activity in lipid peroxidation systems often involves the use of malondialdehyde (MDA) as a commonly employed oxidative stress assay. During the oxidative breakdown of polyunsaturated fatty acids in acidic conditions, MDA is formed. To measure MDA levels, it is combined with Thiobarbituric Acid (TBA) to form the MDA-TBA adduct. This adduct serves as an indicator of lipid peroxidation and is widely utilized to evaluate the effectiveness of various chemicals in mitigating oxidative stress [45,46].

In the present study, four types of samples were tested: blank nanobubbles (NBs), oxygen-loaded NBs, curcumin-loaded NBs, and oxygen-curcumin-loaded NBs. The following procedure was followed for each test tube: 0.1 mL of distilled water, 0.2 mL of sodium dodecyl sulfate (SDS), 1.5 mL of phosphoric acid, and 100 mL of liposomes. Subsequently, 100 µL of the respective sample (NBs) was added to the corresponding test tube.

The mixture was then heated at 100 °C for 45 min after adding 1 mL of the TBA solution (thiobarbituric acid). To cool down the samples, they were placed on an ice bath before adding 4 mL of 1-butanol. Upon addition of 1-butanol, two distinct layers formed, and the supernatant layer containing TBA-MDA adducts was separated. The samples (supernatant layer) were analyzed at 535 nm using a UV spectrophotometer. This procedure allowed for the assessment of the levels of TBA-MDA adducts, providing an indication of the lipid peroxidation in the samples. By measuring the absorbance at 535 nm, the extent of lipid peroxidation could be determined, offering insights into the oxidative stability of the different types of NBs tested.

### 4.7. Statistical Analysis

For every investigational study, at least three independent experiments were executed, and each condition was performed at least in duplicate. Data were expressed as means ± standard errors of the mean (SEM), whereas representative pictures for imaging results were selected. All data were analyzed for significance by Student’s *t*-test (GraphPad Software, version 9, San Diego, CA, USA). A *p* value = 0.0332 was deemed significant.

## 5. Conclusions

Bacterial infections pose a significant threat to human health due to the emergence of antibiotic-resistant strains. Consequently, the scientific community has turned its attention towards natural, green, and sustainable approaches as potential solutions. While these studies utilize natural composites, they are far from simple, employing sophisticated techniques such as bacterial photodynamic inactivation (PDI). PDI can activate natural photosensitizers, such as curcumin, to generate lethal reactive oxygen species (ROS) upon exposure to visible light. Additionally, the development of nanovectors that encapsulate antibacterial substances for sustained and prolonged release in the tissues has emerged as a modern and effective strategy to enhance the antimicrobial fight. These innovative approaches hold promise for combating bacterial infections and addressing the growing concerns of antibiotic resistance.

The objective of the present study was to optimize, synthesize, and evaluate the bactericidal effectiveness of chitosan-shelled and oxygen-loaded nanoformulations containing curcumin. The development and production of Curc-CS-NBs and Curc-Oxy-CS-NBs demonstrated promising physicochemical characteristics and potent antibacterial capabilities. After photoactivation, both types of NBs exhibited increased effectiveness against all tested bacterial strains. In the instance of *E. coli*, Curc-Oxy-CS-NBs displayed a MIC range of 93 µg/mL in the absence of photodynamic activation. However, when NBs were exposed to LED light, the MIC was 46 µg/mL. For *S. aureus*, the MIC in the presence of Curc-Oxy-CS-NBs was 46 µg/mL, without photodynamic activation. Following photodynamic activation, the MIC drastically dropped to 12 µg/mL for Curc-Oxy-CS-NBs. For *E. faecalis*, following LED illumination, the MIC values for Curc-Oxy-CS-NBs decreased to 46µg/mL, starting from an initial 93µg/mL.Inizio moduloFine modulo

Notably, these features were further enhanced by photodynamic activation using visible light (LED). The study demonstrated that a suitable combination of Curc-Oxy-CS-NBs and photodynamic treatment resulted in increased inactivation of the microorganism, including challenging Gram-negative bacteria such as *E. coli*, which are difficult to eliminate due to the distinct characteristics of their cell envelope.

These results further confirm the antibacterial effectiveness of the photoactivated curcumin-loaded NBs which were already proposed for food preservation [6,7], in particular for berries. In this contest, the affordable cost of the materials and the fact that most of the supermarket refrigerated benches are equipped with white light LED, which could, at least partially, elicit photodynamic curcumin activation, makes this approach readily applicable to the market. This innovative approach offers a potential solution to combat bacterial infections, including antibiotic-resistant strains, and holds promise for future applications in the field of antimicrobial therapies.

Additional research into the application of nanobubbles directly on food to combat fungi and bacteria is highly encouraged.

All the components of the curcumin-loaded NBs and of the curcumin-and-oxygen-loaded NBs are of analytical grade and are biocompatible and safe. Their use for food preservation is allowed. Their topical application to infected superficial tissues (skin, oral and vaginal mucosae, etc.) implies that all the requirements for medical devices are satisfied according to the legal constraints of the different countries.

Future studies could focus on optimizing the antimicrobial effects by reducing the curcumin content in the NBs while maintaining their efficiency. Furthermore, investigating shorter photodynamic activation durations while still achieving effective antibacterial activity could be beneficial. Integrating these findings into a comprehensive strategy to combat pathogenic microorganisms would contribute to safeguarding human health and enhancing food quality on a global scale. By continuously refining and advancing these methods, we can better address the challenges posed by foodborne pathogens and ensure safer and healthier food options for consumers worldwide.

## Figures and Tables

**Figure 1 ijms-24-15595-f001:**
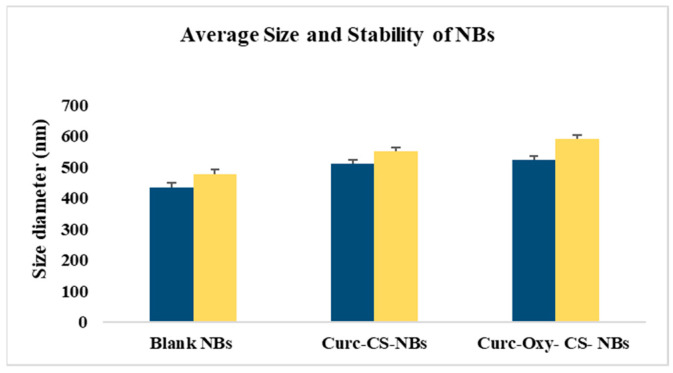
The average diameter of the Curc-CS-NBs, and Curc-Oxy-CS-NBs was measured over time and the results are presented as mean ± standard deviation (SD) based on three samples. The blue color corresponds to measurements at time 0 and the yellow color represents measurements after 2 months.

**Figure 2 ijms-24-15595-f002:**
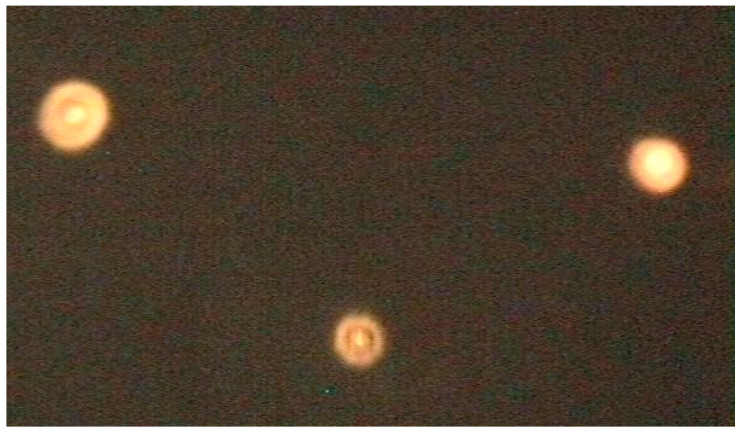
Fluorescence microscopy image depicts the morphology of chitosan-shelled nanobubbles (NBs), with curcumin conjugated to the NBs shell and dissolved in the core (scale bar 500 nm).

**Figure 3 ijms-24-15595-f003:**
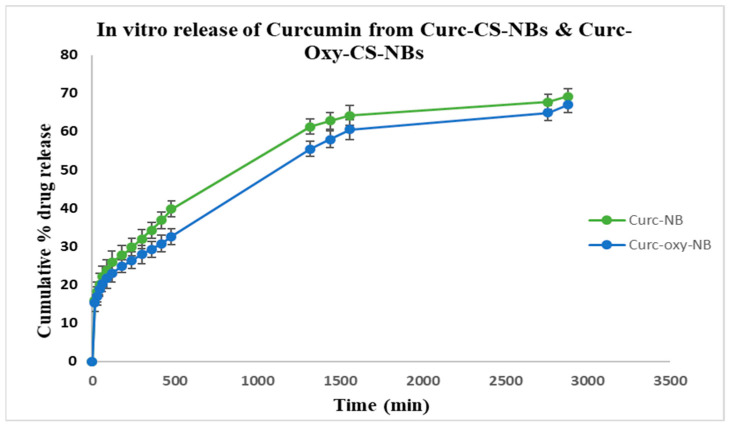
In vitro release kinetics of curcumin from the NBs. The results are reported as the mean ± standard deviation (n = 3). The release profile of Curc-CS-NBs is represented by the green line, while the release profile of Curc-Oxy-CS NBs is represented by the blue line. The in vitro release study, which lasted 48 h, demonstrated the sustained release of curcumin from both the NB formulations over time.

**Figure 4 ijms-24-15595-f004:**
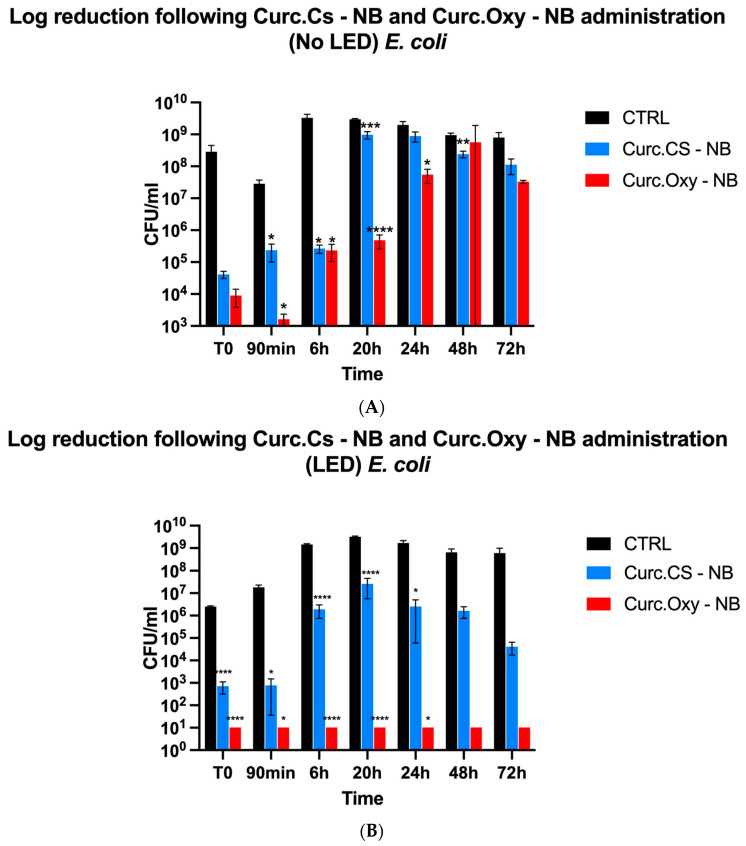
Graphical representation of the killing activity against *E. coli* comparing the efficacy of Curc-NBs and Curc-Oxy-NBs under different conditions. (**A**) Killing activity of Curc-NBs vs. Curc-Oxy-NBs in the absence of LED light. (**B**) Killing activity of Curc-NBs vs. Curc-Oxy-NBs in the presence of LED light. Results are shown as means ± SEM from three independent experiments and expressed as Log CFUs/mL. Data were evaluated for significance by Student’s *t*-test. Vs controls: * *p* = 0.0332; ** *p* = 0.0021; *** *p* = 0.0002; **** *p* < 0.0001.

**Figure 5 ijms-24-15595-f005:**
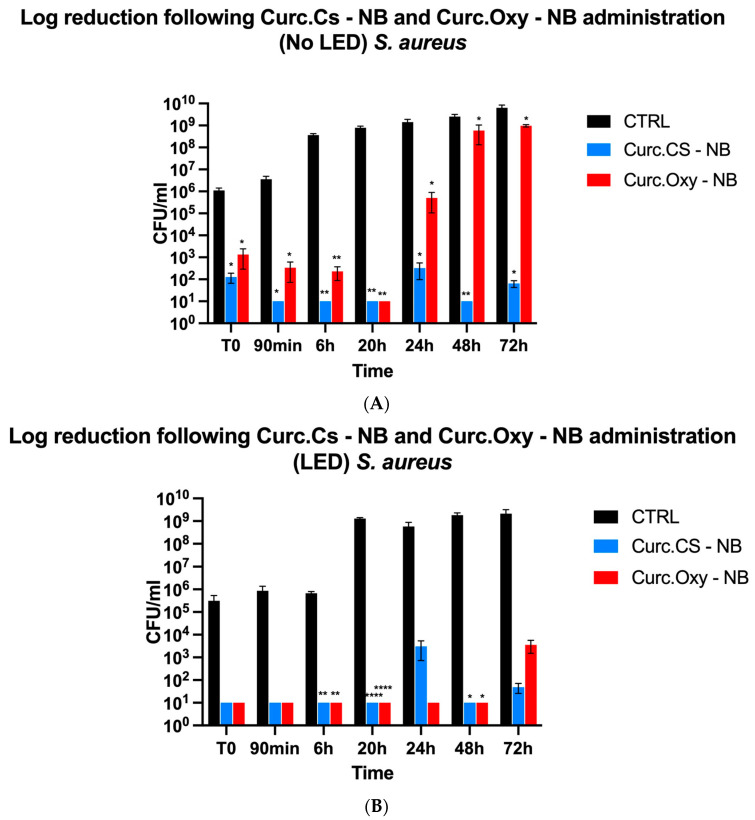
Graphical representation of killing activity against *S. aureus*. (**A**) Comparison between Curc-NBs and Curc-Oxy-NBs without LED light. (**B**) Comparison between Curc-NBs and Curc-Oxy-NBs with LED light. Results are shown as means ± SEM from three independent experiments and expressed as Log CFUs/mL. Data were evaluated for significance by Student’s *t*-test. Vs controls: * *p* = 0.0332; ** *p* = 0.0021; **** *p* < 0.0001.

**Figure 6 ijms-24-15595-f006:**
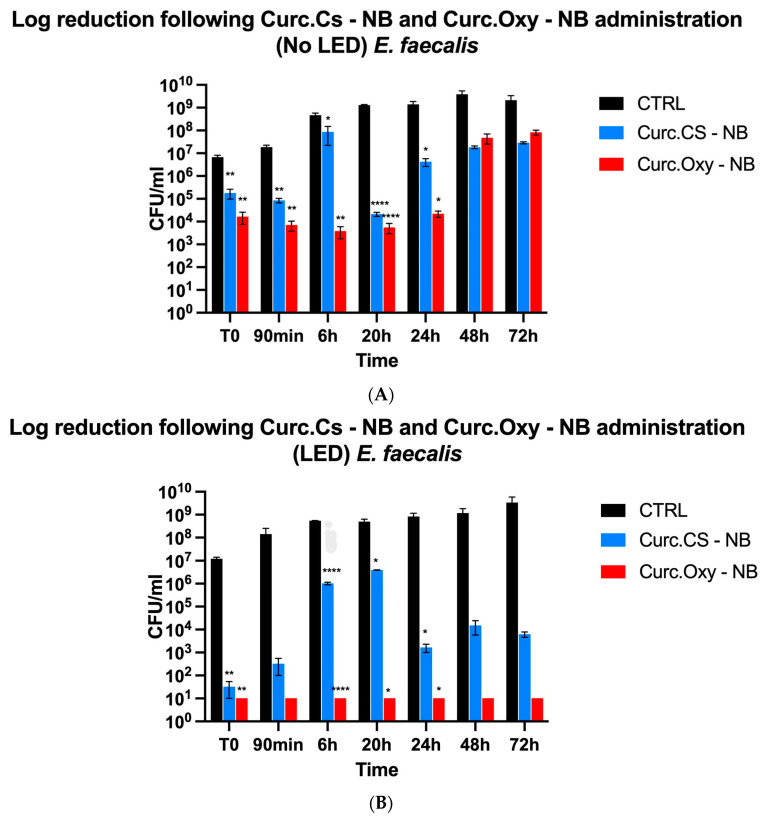
Killing effect against *E. faecalis*. (**A**) Comparison between Curc-NBs and Curc-Oxy-NBs without LED light. (**B**) Comparison between Curc-NBs and Curc-Oxy-NBs with LED light. Results are shown as means ± SEM from three independent experiments and expressed as Log CFUs/mL. Data were evaluated for significance by Student’s *t*-test. Vs controls: * *p* = 0.0332; ** *p* = 0.0021; **** *p* < 0.0001.

**Figure 7 ijms-24-15595-f007:**
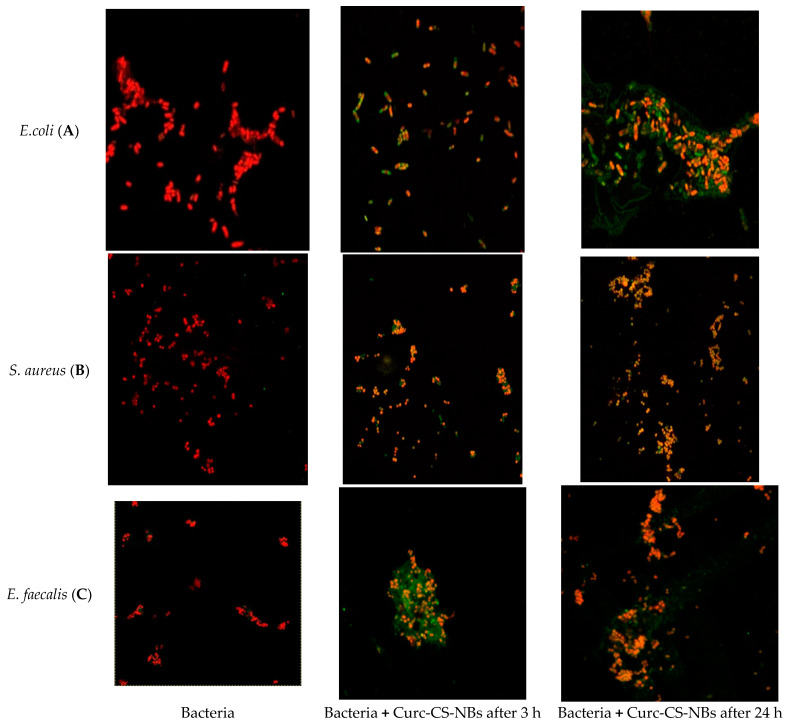
Confocal microscopy images of Curc-CS-NBs’ adhesion to *E. coli* bacterial wall (**A**) and internalization by *S. aureus* (**B**) and *E. faecalis* (**C**). Bacteria (10^9^ CFUs/mL) were left alone or incubated with 10% coumarin-loaded NBs for 3 h and 24 h. Red: PI. Green: coumarin. Magnification: 60×.

**Figure 8 ijms-24-15595-f008:**
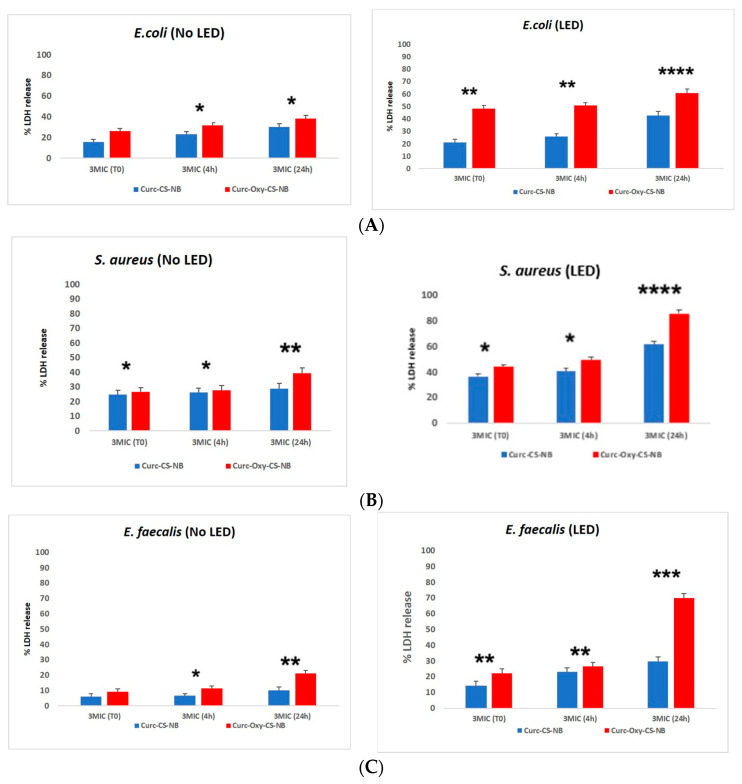
LDH release activity of Curc-CS-NBs and Curc-oxy-CS-NBs under LED and dark condition against the tested bacterial strains. (**A**) LDH for *E. coli*, (**B**) LDH for *S. aureus*, (**C**) LDH for *E. faecalis.* Data were evaluated for significance by Student’s *t*-test. Vs controls: * *p* = 0.0332; ** *p* = 0.0021; *** *p* = 0.0002; **** *p* < 0.0001.

**Figure 9 ijms-24-15595-f009:**
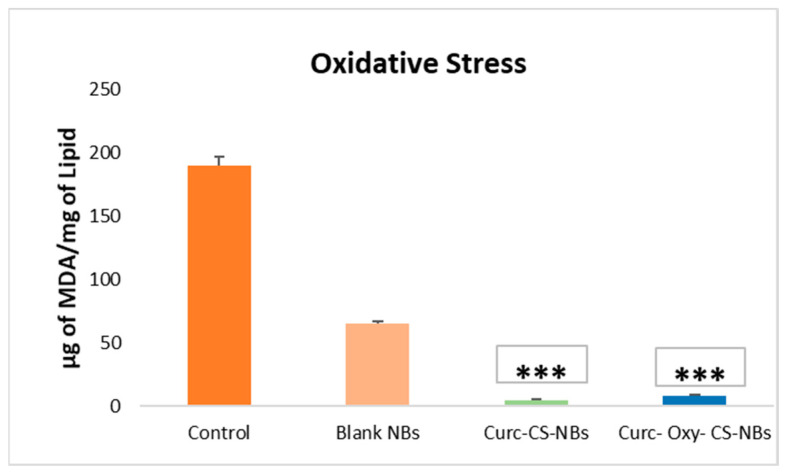
The formulations of Curc-CS-NBs and Curc-Oxy-CS-NBs demonstrate a significant reduction in lipid peroxidation. The statistical analysis shows that the difference is highly significant, denoted by *** indicating *p* < 0.001.

**Figure 10 ijms-24-15595-f010:**
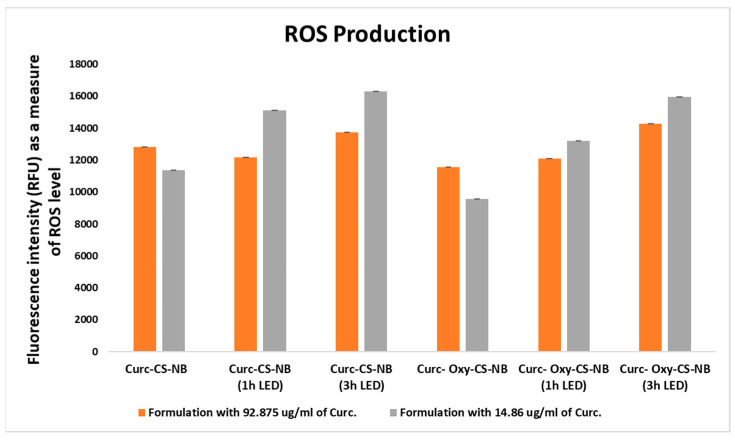
Reactive oxygen species generation with LED irradiation (0–3 h) in response to high and low concentrations of curcumin in nanoformulations (Curc-CS-NB and Curc-CS-Oxy-NB).

**Figure 11 ijms-24-15595-f011:**
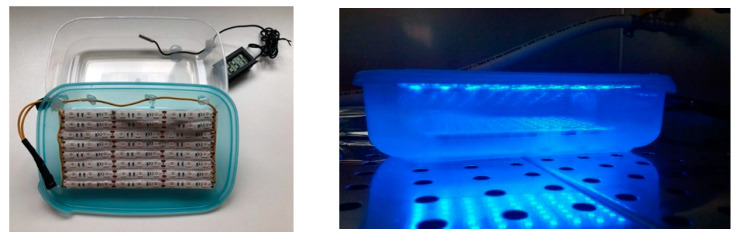
Blue LED (LES Flex Strips LEDYDEL IP64, Turin, Italy).

**Table 1 ijms-24-15595-t001:** The physicochemical characteristics of various formulations of nanobubbles (NBs) were analyzed and the results were reported as mean ± standard deviation (SD).

Formulation	Average Diameter (nm)	Polydispersity Index	Zeta Potential (mV)
Blank NBs	438 ± 18	0.15 ± 0.06	+35 ± 1
Curc-CS-NBs	511 ± 25	0.28 ± 0.02	+30 ± 2
Curc-Oxy-CS-NBs	521 ± 30	0.29 ± 0.02	+31 ± 1

**Table 2 ijms-24-15595-t002:** Encapsulation efficiency and loading capacity of NBs. Mean of the percentages (n = 3) are presented.

Formulation	Encapsulation Efficiency (%)	Loading Capacity (%)	Conc. of Curc. in NBs (µg/mL)
Curc-CS-NBs	87.7	4.3	743 ± 44
Curc-Oxy-CS-NBs	89.2	5.3	743 ± 49

**Table 3 ijms-24-15595-t003:** MIC of Curc-CS-NBs and Curc-Oxy-CS-NBs tested with or without photodynamic treatment on *E. coli*, *S. aureus*, and *E. faecalis*.

Bacteria	MIC (µg/mL) NO LED		MIC (µg/mL) 3 h LED	
	Curc-CS-NBs	Curc-Oxy-CS-NBs	Curc-CS-NBs	Curc-Oxy-CS-NBs
*E. coli*	46.4	92.8	46.4	46.4
*S. aureus*	92.8	46.4	46.4	11.6
*E. faecalis*	46.4	92.8	23.2	46.4

**Table 4 ijms-24-15595-t004:** Composition of the different NB formulations.

Types of NBs	Curcumin in		Oxygen
	Core	Shell	
Chitosan-shelled NB (blank NBs)	No	No	No
Curcumin-loaded and curcumin-conjugated chitosan-shelled NBs (Curc-CS-NBs)	Yes	Yes	No
Curcumin oxygen- and curcumin-loaded conjugated chitosan-shelled NBs (Curc-Oxy-CS-NBs)	Yes	Yes	Yes

## Data Availability

The data presented in this articl are available on request from the Corresponding Author Narcisa Mandras. The data are not publicly available due to privacy.

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
