# Peer review of "Encapsulation in Oxygen-Loaded Nanobubbles Enhances the Antimicrobial Effectiveness of Photoactivated Curcumin"

_ijms, 2023, doi:10.3390/ijms242115595_

Round 1
Reviewer 1 Report
The paper presents an interesting and potentially impactful approach to combat antibiotic resistance in healthcare and agriculture by utilizing nanobubbles (NBs) to enhance the antibacterial properties of curcumin. The authors highlight the importance of biocompatible and biodegradable ingredients in addressing the growing concern of antibiotic resistance, aligning with the "green" criteria advocated by the Next Generation EU platform.
The use of innovative strategies like nanoformulations and photoactivation to improve curcumin's solubility, stability, and antibacterial effectiveness is a noteworthy aspect of this study. The concept of encapsulating curcumin in nanobubbles and utilizing it as a natural photosensitizer for photodynamic stimulation is innovative and holds promise.
Key strengths of the paper include:
Relevance: The paper addresses a pressing issue of antibiotic resistance in both healthcare and agriculture, highlighting the need for sustainable and effective solutions. The focus on curcumin, a natural compound with known antimicrobial properties, adds relevance to the study.
Innovation: The use of nanobubbles to encapsulate curcumin and the application of photodynamic stimulation to enhance its antibacterial efficacy represent innovative approaches that can contribute to the development of eco-friendly substitutes.
Comprehensive Approach: The study employs a comprehensive approach by investigating both Curc-CS-NBs and Curc-Oxy-CS-NBs, providing a thorough analysis of their antibacterial properties. The inclusion of Gram-positive and Gram-negative bacteria adds depth to the research.
Potential Implications: The findings indicate promising results in terms of bacterial elimination, including challenging Gram-negative bacteria. This suggests the potential of the proposed strategy to combat antibiotic resistance effectively.
However, there are some aspects that could be further elaborated or addressed in the paper:
Methodology and Experimental Design: A more detailed description of the methodology and experimental design would be beneficial for readers seeking to replicate or build upon this research. This includes information on nanobubble synthesis, characterization, and the specifics of the photoactivation process.
Data Presentation: The paper briefly mentions the outcomes, but it would be more informative if it included quantitative results and statistical analysis to support the claims of effectiveness against different bacteria.
Safety and Biocompatibility: Since the study discusses potential applications in healthcare, it is important to address safety and biocompatibility concerns associated with the proposed nanobubble-based approach.
Future Directions: It would be valuable to discuss potential future directions for this research, such as scalability, cost-effectiveness, and real-world applications in healthcare and agriculture.
In addition, some important reference should be cited. Trends in Chemistry, vol. 4, no. 12, pp. 1065 – 1077 (2022). Rare Met. (2021) 40(9):2447–2454
good
Author Response
ANSWER TO REVIEWER 1 COMMENTS
- Methodology and Experimental Design: A more detailed description of the methodology and experimental design would be beneficial for readers seeking to replicate or build upon this research. This includes information on nanobubble synthesis, characterization, and the specifics of the photoactivation process.
Please see the revised version. In Materials and Methods a more detailed description of methodology and experimental desing were added. The revised part is marked with yellow highlight.
- Data Presentation: The paper briefly mentions the outcomes, but it would be more informative if it included quantitative results and statistical analysis to support the claims of effectiveness against different bacteria.
Thank you very much for the professional suggestion. The results were revised as suggested. We eliminated Figures 4-7 and added Table 3 and Figures 4-6 and have included statistical analysis. The revised part is marked with yellow highlight.
- Safety and Biocompatibility: Since the study discusses potential applications in healthcare, it is important to address safety and biocompatibility concerns associated with the proposed nanobubble-based approach.
We agree. Please see the revised version. Please see at page 22.
- Future Directions: It would be valuable to discuss potential future directions for this research, such as scalability, cost-effectiveness, and real-world applications in healthcare and agriculture
Following your suggestion, we added this part. Please see at page 22
Reviewer 2 Report
-Please briefly state your inhibition zones for antibacterial analyzes in the abstract section.
-There is not enough related work analysis data in the introduction section. Please explain in detail the outstanding promise of this research by stating the analysis data of the literatures. The following research will help you
Biocidal Activity of Bone Cements Containing Curcumin and Pegylated Quaternary Polyethylenimine
T Eren, G Baysal, F DoÄŸan
Journal of Polymers and the Environment 28, 2469-2480
-Please explain what the images obtained in morphological studies mean.
-please refer to the equations and equations used in the manuscript (encapsulation efficiency, EE and LC).
-Please include a table comparing all results for microbial analyzes with this study and the literature.
-Try to present the research more strikingly by sharing the numerical data of this study in the conclusion section. This is quite impressive research. Congratulations
-
Author Response
ANSWER TO REVIEWER 2 COMMENTS
- Please briefly state your inhibition zones for antibacterial analyzes in the abstract section.
Thank you very much for the suggestion. However, we have no inhibition zone data but following your suggestion the Abstract were revised remembering that it should be a total of about 200 words maximum.
2. There is not enough related work analysis data in the introduction section. Please explain in detail the outstanding promise of this research by stating the analysis data of the literatures. The following research will help you
Biocidal Activity of Bone Cements Containing Curcumin and Pegylated Quaternary Polyethylenimine, T Eren, G Baysal, F DoÄŸan Journal of Polymers and the Environment 28, 2469-2480
Thank you for the valuable comment. We have added some references including the suggested work. The revised part is marked with yellow highlight.
3. Please explain what the images obtained in morphological studies mean.
Done. Please see page 4, Figure 2
- Please refer to the equations and equations used in the manuscript (encapsulation efficiency, EE and LC).
Following your suggestion, the results were revised. Please see page 4 Paragraph 2.3
- Please include a table comparing all results for microbial analyzes with this study and the literature.
Thank you very much for the suggestion. However we preferred to include in the discussion the few comparisons we found in the literature. Please see the Discussion section.
- Try to present the research more strikingly by sharing the numerical data of this study in the conclusion section. This is quite impressive research. Congratulations
Following your suggestion the Conclusion section were revised as suggested.